# Differential Impact of IL-32 Isoforms on the Functions of Coronary Artery Endothelial Cells: A Potential Link with Arterial Stiffness and Atherosclerosis

**DOI:** 10.3390/v15030700

**Published:** 2023-03-08

**Authors:** Rémi Bunet, Marie-Hélène Roy-Cardinal, Hardik Ramani, Aurélie Cleret-Buhot, Madeleine Durand, Carl Chartrand-Lefebvre, Jean-Pierre Routy, Réjean Thomas, Benoît Trottier, Petronela Ancuta, David B. Hanna, Alan L. Landay, Guy Cloutier, Cécile L. Tremblay, Mohamed El-Far

**Affiliations:** 1Département de Microbiologie, Infectiologie et Immunologie, Faculté de Médecine, Université de Montréal, Montréal, QC H3C 3J7, Canada; 2Centre de Recherche du Centre Hospitalier de l’Université de Montréal (CRCHUM), Montréal, QC H2X 0A9, Canada; 3Laboratory of Biorheology and Medical Ultrasonics, Centre de Recherche du Centre Hospitalier de l’Université de Montréal (CRCHUM), Montréal, QC H2X 0A9, Canada; 4Département de Médecine, Faculté de Médecine, Université de Montréal, Montréal, QC H3T 1J4, Canada; 5Département de Radiologie, Radio-Oncologie et Médecine Nucléaire, Faculté de Médecine, Université de Montréal, Montréal, QC H3C 3J7, Canada; 6Chronic Viral Illness Service, Division of Hematology, McGill University Health Centre, Montréal and Research Institute of McGill University Health Centre, Montréal, QC H4A 3J1, Canada; 7Clinique Médicale l’Actuel, Montréal, QC H2L 4P9, Canada; 8Centre de Médecine Urbaine du Quartier Latin, Montréal, QC H2L 4E9, Canada; 9Department of Epidemiology and Population Health, Albert Einstein College of Medicine, Bronx, NY 10461, USA; 10Department of Internal Medicine, Rush University Medical Center, Chicago, IL 60612, USA; 11Institut de Génie Biomédical, Université de Montréal, Montréal, QC H3T 1J4, Canada

**Keywords:** HIV, inflammation, cardiovascular disease, IL-32, arterial stiffness, endothelial cell dysfunction, CCL-2, CXCL-8, CXCL-1

## Abstract

Chronic inflammation is associated with higher risk of cardiovascular disease (CVD) in people living with HIV (PLWH). We have previously shown that interleukin-32 (IL-32), a multi-isoform proinflammatory cytokine, is chronically upregulated in PLWH and is linked with CVD. However, the mechanistic roles of the different IL-32 isoforms in CVD are yet to be identified. In this study, we aimed to investigate the potential impact of IL-32 isoforms on coronary artery endothelial cells (CAEC), whose dysfunction represents a major factor for atherosclerosis. Our results demonstrated that the predominantly expressed IL-32 isoforms (IL-32β and IL-32γ) have a selective impact on the production of the proinflammatory cytokine IL-6 by CAEC. Furthermore, these two isoforms induced endothelial cell dysfunction by upregulating the expression of the adhesion molecules ICAM-I and VCAM-I and the chemoattractants CCL-2, CXCL-8 and CXCL-1. IL-32-mediated expression of these chemokines was sufficient to drive monocyte transmigration in vitro. Finally, we demonstrate that IL-32 expression in both PLWH and controls correlates with the carotid artery stiffness, measured by the cumulated lateral translation. These results suggest a role for IL-32-mediated endothelial cell dysfunction in dysregulation of the blood vessel wall and that IL-32 may represent a therapeutic target to prevent CVD in PLWH.

## 1. Introduction

Life expectancy of people living with HIV (PLWH) has been greatly enhanced with anti-retroviral therapy, ART [1,2]. However, long-term health management is strictly needed as PLWH are aging and are exposed to age-associated comorbidities. For instance, 10% of PLWH worldwide are now over the age of 50, whereas in US and Canada, the aging population represents 50% of PLWH [3]. This natural biological aging is further associated with inflammation and chronic activation of the immune system in PLWH even under ART, which accelerates the development of multiple comorbidities such as cancer, neurological, and cardiovascular diseases [4,5,6]. In this regard, arterial dysfunction is known to be involved in cardiovascular diseases including hypertension, stroke and heart disease [7,8]. The development of dysfunctional arterial endothelium with age contributes to a number of hemodynamic changes in the body by increasing large artery stiffness, oscillatory shear stress and resistant arterial tone [9].

At the cellular level, endothelial dysfunction leads to an increase in CXCL-8 and CCL-2 chemokine secretion and increased expression of the adhesion molecules ICAM-I and VCAM-I [10,11]. Among these, CCL-2 is a strong chemoattractant that guides monocyte trafficking to the inflamed arteries and combined with CXCL-8 they play a role in firm adhesion of monocyte into the endothelium [12]. Meanwhile, upregulation of the adhesion molecules ICAM-I and VCAM-I promotes slow rolling, firm adhesion and migration of monocytes through the endothelium [13]. This endothelial dysfunction phenotype increases extravasation and subendothelial accumulation of monocytes that can differentiate into macrophages and give rise to the pathogenic foam cells [14,15]. Along with the endothelial dysfunction, elevated levels of IL-1β, IL-6 and TNF-α inflammatory cytokines contribute to atherosclerotic plaque formation and cardiac irritability [10,16,17]. IL-6 and TNF-α have been directly correlated with obstructive coronary artery disease (CAD) along with IL-32 that was shown to be highly expressed in the coronary artery endothelium in people with CAD [18,19].

IL-32 is a pro-inflammatory cytokine produced by immune and nonimmune cells and is expressed in at least 10 different isoforms (α, β, γ, δ, D, ε, θ, ζ, η, and small/sm, generated by alternative splicing) [20,21,22]. We and others have shown that plasmatic IL-32 levels are upregulated in PLWH even under ART treatment [23,24]. We further demonstrated that specific IL-32 isoforms (IL-32 α, β, and ε in women but IL-32 D and θ in men living with HIV) were associated with carotid artery and coronary artery subclinical atherosclerosis, respectively [25,26]. These observations warranted further studies to dissect the roles of IL-32 isoforms in mechanisms underlying the development of CVD in PLWH. Herein, we aimed to investigate the impact of exogenous IL-32 isoforms on the dysfunction of coronary artery endothelial cells (CAEC) and carotid artery stiffening, and to identify their profile of key cytokines and chemokines secretion in response to IL-32 activation.

## 2. Materials and Methods

### 2.1. The Study Participants

The current study included PLWH (n = 60) and control (n = 53) individuals participating in the Canadian HIV and Aging Cohort Study (CHACS). The CHACS cohort follows longitudinally PLWH and controls for the development of cardiovascular diseases where subclinical atherosclerosis is measured using a cardiac computed tomography (CT) scan with injection of contrast media and defined by the presence of atherosclerotic plaque (plaque+) in the coronary arteries [27].

### 2.2. Cells

Primary Human Coronary Artery Endothelial Cells were obtained from Creative Bioarray (Shirley, NY, USA). Cells were maintained in culture in Human Coronary Artery Endothelial Cell Medium (Creative Bioarray, Shirley, NY, USA) and experiments were conducted between passage 5 to 10.

### 2.3. Flow Cytometry Analysis

Flow cytometry analysis was used to study the expression of adhesion molecules on CAEC using BD FACSAria (BD Biosciences, San Jose, CA, USA). Staining was performed with 1 million CAEC cells detached with Accutase (STEMCELL, Vancouver, CAN) in 4% Fetal Bovine Serum (FBS) in PBS. Cell surface staining was performed with fluorochrome-conjugated antibodies from Biolegend (San Diego, CA, USA); Alexa Fluor 700 Mouse anti-Human CD54 (Clone HA58, Cat # 353126), APC Mouse anti-Human CD106 (Clone STA, Cat # 305810) and FITC Mouse anti-Human CD31 (Clone WM59, Cat # 303104).

### 2.4. Cell Stimulation

CAEC were stimulated with 500 ng/mL of IL-32 isoforms α (RND Systems Minneapolis, MN, USA Cat # 3040-IL-050), β (Cat # 6769-IL-025) and γ (Cat # 4690-IL-025/CF) at 90% cell confluence. Cells were incubated for 3 h, 12 h or 72 h at 37 °C with 5%CO_2_ as appropriate and as described in the Figure legends. All stimulations were done in the presence of Polymyxin B (inhibitor of the lipopolysaccharide LPS) at 1 μg/mL.

### 2.5. ELISA

IL-10, IL-18, IL-6, TNF-α, IL-1β, CCL-2, CXCL-1, CXCL-8, ICAM-I and VCAM-I were quantified using commercially available ELISA kits (R&D Systems, Minneapolis, MN, USA Cat # DY217B-05, DY318-05, DY406-05, DY210-05, DY201-05, DY279-05, DY275-05, DY208-05, DY720-05 and DY809-05, respectively) as per the manufacturer’s recommendations in CAEC supernatant.

### 2.6. Quantitative Reverse Transcription Assays (RT-qPCR)

Total RNA was isolated from CAEC using the RNeasy plus mini kit (QIAGEN, Toronto, ON, CAN. Catalog No. 74134) as per the manufacturer’s protocol. Primers sets for the quantification of the different genes were as follows: CXCL-1 (Forward 5′- AACCGAAGTCATAGCCACAC-3′and reverse 5′- GTTGGATTTGTCACTGTTCAGC-3′) [28], CXCL-8 (Forward 5′- GACCACACTGCGCCAACAC-3′ and reverse 5′- CTTCTCCACAACCCTCTGCAC-3′) [29], CCL-2 (Forward 5′-CAGCCAGATGCAATCAATGC-3′ and reverse 5′- TGGAATCCTGAACCCACTTCT-3′) [30], ICAM-I (Forward 5′-GGCCGGCCAGCTTATACA-3′ and reverse 5′-TAGACACTTGAGCTCGGGCA-3′) [31] and VCAM-I (Forward 5′-TCAGATTGGAGACTCAGTCATG-3′ and reverse 5′-ACTCCTCACCTTCCCGCTC-3′) [31]. Primers used for the IL-32 isoforms quantification can be found in Appendix A in our previous publication [22]. The IL-32 isoforms quantification assays were carried out using one-step SYBR Green quantitative real-time PCR on RNA from PBMCs isolated from the study participants and extracted using the RNeasy plus mini kit from Qiagen as per the manufacturer’s protocol. RT-qPCR was performed on a LightCycler 480 II (Roche, Mississauga, ON, CAN) apparatus using QIAGEN (Toronto, ON, CAN) reagents (RNase-Free DNase Set (50); Cat. No. 79254). Real-time RT-qPCR was performed using 25ng RNA per reaction. Gene expression was normalized to the housekeeping gene β-glucuronidase. RT-qPCR data were analyzed with the ΔCT method using the housekeeping gene β-glucuronidase as an internal reference as we previously described [22].

### 2.7. Transwell Assay

Negatively selected monocyte (STEMCELL, Vancouver, BC, CAN Cat # 19058) migration was carried out in a 3 μm transwell insert (CELLTREAT, Pepperell, MA, USA. Cat #230631) by placing the monocytes in the upper chamber while placing the supernatants from IL-32 isoform-stimulated CAEC in the lower chamber for 3 h with or without antagonists for CXCR2 (TOCRIS, Toronto, ON, CAN. Cat # 2725) and CCR2 (TOCRIS, Toronto, CAN. Cat # 2517). Transwell membranes were stained with DAPI (Sigma, Saint-louis, MO, USA Cat # D9542) for cell nucleus and imaged with a Zeiss microscope with Z-stack. Automatic cell counting was performed using FIJI (NIH, ImageJ2, v1.8.0_172) macro developed by the cell imaging facility of the CRCHUM.

### 2.8. Carotid Artery Ultrasound Imaging and Image Analysis

Ultrasound imaging and image analysis have been carried as described in our previous work [32]. Ultrasound acquisitions were performed between October 2015 and October 2017. Arterial walls of left and right common and internal carotids were assessed in this study; 4 cine-loops were thus acquired for each participant. Longitudinal views of each vessel were acquired with an Aixplorer system (SuperSonic Imagine, Aix-en-provence, FRA) using a 256-element linear array probe (SuperLinear™ SL15-4) at 7.5 MHz. Participants were asked to hold breath during acquisitions. The frame rate was set to 50 frames per s, and cine-loops of raw US radiofrequency data were recorded for approximately 5 s. Examinations were performed by a vascular technologist with more than 20 years of experience.

### 2.9. Statistical Analysis 

Data were analyzed using GraphPad Prism 8 (GraphPad software, San Diego, CA, USA) and SAS 9.4 (SAS Institute, Cary, NC, USA). Non-parametric Kruskal–Wallis and Dunn’s subtests were used to analyze more than 2 groups for the same variable whereas Mann-Whitney non-parametric analysis was used to compare two groups for the same variable. Linear regression was used for multivariable analysis, adjusting for age, Framingham risk score, body mass index, and LDL and HDL cholesterol levels. Differences between groups were considered statistically significant at values of *p* < 0.05 with two-tailed analysis.

## 3. Results

### 3.1. IL-32 Isoforms Exhibit a Differential Impact on Cytokine Production by the Primary Coronary Artery Endothelial Cells

We first aimed to determine whether IL-32 isoforms would upregulate the production of inflammatory cytokines by the primary CAEC based on our previous observations on both CD4 T-cells and monocytes where IL-32 isoforms induce multiple inflammatory mediators [24]. We tested the production of key cytokines involved in CVD such as IL-6, TNF-α, IL-1β and IL-18 as well as the anti-inflammatory cytokine IL-10 in CAEC supernatant by ELISA following 72 h of stimulation with IL-32 isoforms. We observed that IL-32β and γ, but not IL-32α, significantly upregulated the production of the inflammatory cytokine IL-6 (*p* < 0.0001 and *p* = 0.0095, respectively) compared to the non-stimulated conditions (Figure 1A, left panel). In contrast to what we observed on other cell types [24], CAEC did not upregulate any other inflammatory cytokines such as TNF-α, IL-1β or IL-18 in response to IL-32 isoforms (Figure 1B), or even to LPS stimulation (data not shown), suggesting that CAEC are not a source for these inflammatory cytokines. Interestingly, we observed that IL-32β, and to a lesser extent IL-32γ, decreased the production of the anti-inflammatory cytokine IL-10 in CAEC (Figure 1A, right panel, *p* = 0.0011 for IL-32β). These results demonstrated that, in contrast to the impact of IL-32 isoforms on T-cells and monocytes where they induce a number of inflammatory cytokines [22,24,26], endothelial cells respond to IL-32 isoforms by increasing the expression of the CVD-associated cytokine IL-6 and decreasing the atheroprotective cytokine IL-10.

### 3.2. IL-32 β and γ Induce Coronary Artery Endothelial Cell Dysfunction

Given the prominent role of IL-6 in endothelial cell dysfunction [33] and the specific upregulation of IL-6 by CAEC in response to IL-32 as shown above, we aimed to determine the impact of the IL-32 isoforms on the typical endothelial dysfunction markers ICAM-I and VCAM-I [34]. We quantified these adhesion molecules at the RNA level by RT-qPCR in CAEC following a 12h stimulation with IL-32 isoforms α, β or γ. We observed that IL-32β and γ significantly upregulated ICAM-I (*p* = 0.0028 and *p* = 0.0303, respectively) and VCAM-I (*p* = 0.0016 and *p* = 0.0383, respectively) RNA expression compared to non-stimulated cells (Figure 2A). The upregulated RNA expression of ICAM-I and VCAM-I was associated with significantly increased levels of their secreted proteins quantified by ELISA in the supernatants of CAEC following 72 h stimulation with IL-32β and γ (ICAM-I; *p* = 0.0053 and *p* = 0.0068, respectively, and VCAM-I; *p* < 0.0001 and *p* = 0.0001, respectively, Figure 2B). Furthermore, the expression of ICAM-I and VCAM-I on the cell surface (a feature associated with the recruitment of leukocytes to the inflammatory sites) was assessed by flowcytometry following 72 h stimulation under the same conditions. In line with the RNA transcription and secreted protein data, IL-32β and γ significantly upregulated ICAM-I (*p* < 0.0001 and *p* = 0.0005, respectively) and VCAM-I (*p* < 0.0001 and *p* = 0.0007, respectively) on the surface of CAEC compared to non-stimulated cells (Figure 2C). Taken together, these results suggest that IL-32 isoforms β and γ, but not α, induce cell dysfunction in CAEC.

### 3.3. IL-32 β and γ Increase Chemokine Production in Coronary Artery Endothelial Cells

Expression of the adhesion molecules ICAM-I and VCAM-I, which are involved in leukocyte recruitment by endothelial cells, further suggests that IL-32 isoforms might also impact the chemokine production in CAEC, which together (adhesion molecules and chemokines) would enhance the leukocyte recruitment potential of these cells to the site of inflammation. To this end, we evaluated the expression of key chemokines known to be involved in leukocytes chemotaxis, namely, CCL-2, CXCL-1, and CXCL-8 [35,36,37,38]. These chemokines were first quantified by RT-qPCR following a 12 h stimulation with IL-32 isoforms. Similar to the CAEC dysfunction markers, we observed that IL-32β and γ significantly upregulated the RNA expression of CCL-2 (*p* < 0.0001 and *p* = 0.0012 respectively), CXCL-1 (*p* = 0.0001 and *p* = 0.0023 respectively) and CXCL-8 (*p* < 0.0001 and *p* = 0.0028, respectively) compared to the non-stimulated condition or IL-32α-stimulated cells (Figure 3A). The protein expression of these chemokines was further assessed in CAEC supernatant by ELISA following 72 h stimulation with IL-32 isoforms. Of note, all experiments were carried out in the presence of the endotoxin inhibitor Polymyxin B, which did not significantly impact the CAEC response to IL-32 isoforms, while completely diminishing the impact of LPS stimulation that was used at a similar concentration to IL-32 isoforms (500 ng/mL) (Appendix A). Under these experimental conditions, we observed that IL-32β and γ upregulated CCL-2 (*p* < 0.0001 and *p* = 0.0132, respectively), CXCL-1 (*p* < 0.0001 and *p* = 0.0198, respectively) and CXCL-8 (*p* = 0.030 and *p* = 0.066 (marginally significant), respectively) compared to the non-stimulated or IL-32α- stimulated cells (Figure 3B). These results suggested a specific effect of IL-32β and γ to induce chemokines’ expression by CAEC, which would potentially increase their capacity to attract immune cells.

### 3.4. IL-32 β- and γ-Induced Chemokines Drive Monocyte Transmigration towards CAEC Supernatants

Since CCL-2 and CXCL-8 are known to be potent monocyte chemoattractants [36], we hypothesized that CAEC stimulated with IL-32 isoforms would increase their potential to recruit monocytes. We assessed the transmigration of primary monocytes towards the supernatant of CAEC stimulated with IL-32 isoforms α, β and γ in transwell assays and the total migrated cells were counted by imaging (DAPI stain was used to identify the migrated cells on the transwell membranes). We observed that supernatant from IL-32β- and γ-stimulated CAEC significantly attracted higher numbers of monocytes (*p* = 0.0005 and *p* = 0.0141, respectively) compared to the non-stimulated cells (Figure 4A). To ensure this effect was specific and related to the upregulation of CCL-2 and CXCL-8 by IL-32 isoforms, we pre-treated the monocytes with the antagonists RS504393 and SB225002, which target CCR2 (CCL-2 receptor) and CXCR2 (CXCL-8 receptor), respectively, before running the transwell assay. Under these conditions, we observed a significant decrease in the number of transmigrating monocytes compared to the non-treated cells (IL-32β: *p* = 0.0018 for CXCR2 antagonist and *p* = 0.0051 for CCR2 antagonist, IL-32γ: *p* = 0.0045 for CXCR2 antagonist and *p* = 0.0078 for CCR2 antagonist, Figure 4B). Of note, the use of CXCR2 antagonist showed a modest but significant impact on the non-stimulated and IL-32α-stimulated supernatants (*p* = 0.0039 and *p* = 0.0003, respectively), likely driven by the inhibition of the spontaneous CXCL-8 production by CAEC as could be seen in Figure 3B. Intriguingly, the use of either CCR2 or CXCR2 antagonists in the IL-32-stimulated conditions strongly diminished the monocyte migration to levels close to the non-stimulated cells. However, earlier studies demonstrated that monocyte migration by these chemokines (CCL-2 and CXCL-8) and chemokine receptors (CCR2 and CXCR2) are highly interdependent through a synergistic mechanism involving the activation of ERK1/ERK2 pathways and intracellular calcium signaling [39]. These studies demonstrated that CXCL-8 largely enhances the chemoattraction potential of CCL-2 [39]. Therefore, inhibition of CCL-2 in our assays would likely diminish the monocytes migration as expected, but additionally, CXCL-8 inhibition would, on its turn, diminish the synergistic effect for CCL-2-mediated migration as well. Taken together, these results suggest that IL-32-induced expression of the chemokines CCL-2 and CXCL-8 has the potential to significatively and specifically increase monocyte recruitment by coronary artery endothelial cells.

### 3.5. IL-32 Expression Is Associated with Carotid Artery Wall Stiffness

Since we had established the functional link between IL-32 and endothelial cell’s dysfunction and cytokine/chemokine production, we further aimed to investigate whether IL-32 expression in vivo might be correlated with arterial diseases. Accordingly, we quantified the IL-32 isoforms α, β, γ, D, ε and θ (isoforms for which we have established RT-qPCR assays [22,25,26]) from both PLWH (n = 60) and controls (n = 53) (Table 1), and studied their correlation with arterial stiffness determined by cumulated lateral translation in the common carotid artery (CLT-CCA) that is measured by ultrasound noninvasive vascular elastography (NIVE) [32]. CLT-CCA represents the cyclic translation motion, during pulsation, of the arterial wall along the longitudinal vessel axis. Consistent with our earlier observations [22,25,26], all IL-32 isoforms that were tested (α, β, γ, D, ε and θ) were significantly upregulated in PBMCs from PLWH compared to controls (*p* < 0.0001, *p* < 0.0001, *p* < 0.0001, *p* = 0.0049, *p* = 0.0001 and *p* < 0.0001, respectively) (Figure 5A). Interestingly, the levels of IL-32 isoforms (α, β, γ, D, ε and θ) negatively and significantly correlated with cumulated lateral translation in the common carotid artery (CLT-CCA) from the controls (*p* = 0.042, *p* = 0.017, *p* = 0.018, *p* = 0.019, *p* = 0.009 and *p* = 0.03, respectively, Figure 5B). These associations remained statistically significant after adjustment for age, Framingham risk score, body mass index, and LDL and HDL cholesterol levels, with the exception of isoforms α and D which became marginally significant (*p* = 0.09 and *p* = 0.06, respectively). In PLWH, no significant correlation was observed between IL-32 and CLT-CCA when we analyzed the total population of PLWH (Appendix A). However, the negative significant correlations with IL-32 isoforms in PLWH were only observed in the CLT-CCA measures within the upper quartiles (n = 30) but not the lower quartiles (n = 30) (*p* = 0.015, *p* = 0.029, *p* = 0.024 for IL-32γ, D and ε, respectively, and marginally significant for isoforms α, β and θ with *p* = 0.054, *p* = 0.071 and *p* = 0.061, respectively) (Figure 5C). In further analyses adjusted for age, Framingham risk score, body mass index, and LDL and HDL cholesterol levels, the associations between IL-32 and CLT-CCA strengthened and were statistically significant for all isoforms except for θ (*p* = 0.09). These results were intriguing given the higher levels of IL-32 isoform expression in PLWH compared to the controls. However, when we stratified PLWH by age for the upper and lower CLT-CCA quartiles, we observed that individuals within the lower quartile of CLT-CCA (more arterial stiffness) were older compared to individuals within the upper CLT-CCA quartiles (individuals with less diseased arteries) (Appendix A, left panel). Of note, this effect was not observed within the controls, where individuals within the upper (n = 27) and lower (n = 26) CLT-CCA quartiles had similar ages (Appendix A, left panel). These results demonstrated that the association between IL-32 expression and arterial stiffness was only observed in the relatively younger but not the older PLWH, even with the significant correlation between IL-32 and aging (Appendix A, right panel). These results also suggested that IL-32 may accelerate co-morbidities such as vascular diseases in PLWH at an earlier age; however, advanced biological aging remains a dominant mediator/predictor for the deterioration of the arterial functions.

## 4. Discussion

IL-32 is known to be upregulated in multiple inflammatory conditions associated with increased risk for the development of cardiovascular diseases, such with chronic viral and bacterial infections, inflammatory bowel disease and chronic obstructive pulmonary disease [40,41,42,43]. Previous studies including ours demonstrated that IL-32 is persistently upregulated in both cells and plasma from people living with HIV and is associated with the presence of coronary artery and carotid artery atherosclerosis [23,25,26,44]. However, whether IL-32 contributes to the pathogenesis of atherosclerosis is not clear. In the current study, we assessed the impact of IL-32 isoforms on endothelial cell functions and cytokines/chemokines expression as it relates to inflammation and recruitment of leukocytes. We demonstrated that specific exogenous IL-32 isoforms (IL-32β and γ) have the potential to impact coronary artery endothelial cells leading to their dysfunction and recruitment of monocytes, two conditions linked with the development of atherosclerosis [45,46]. We first assessed CAEC pro-inflammatory cytokine production for IL-6, TNF-α, IL-18 and IL-1β, which are four important players for the development of cardiovascular diseases [47,48,49] together with the anti-inflammatory cytokine IL-10. Interestingly, only IL-6 was upregulated by IL-32β and γ as these CAEC do not produce either TNF-α, IL-18 or IL-1β, even in response to LPS. IL-6 is considered to be an upstream inflammatory cytokine that is associated with endothelial dysfunction (mediating the upregulation of the leukocytes’ adhesion molecules ICAM-I and VCAM-I) and subclinical atherosclerosis [33,50] and therefore, its production by CAEC in response to IL-32 β and γ may represent one of the mechanisms by which IL-32 contributes to coronary artery dysfunction and inflammation. To validate this hypothesis, we investigated the dysfunction of CAEC following IL-32 β and γ stimulation by measuring their expression of ICAM-I and VCAM-I. ICAM-I and VCAM-I were both upregulated at the RNA level as well as at the protein level on cell surface. ICAM-I and VCAM-I have been shown to be removed from the endothelial cell surface by proteolytic cleavage and shedding as a control mechanism to limit the effect of the inflammatory process [51] and high levels of plasmatic ICAM-I and VCAM-I have been shown to directly correlate with atherosclerosis [52]. This phenotype was also induced by IL-32 in the current study and indicated that IL-32β and γ induce the upregulation of typical dysfunction biomarkers in CAEC. However, it remains unclear whether this mechanism is directly mediated by IL-6 and how IL-6 may signal in the context of endothelial cells in vitro.

On top of being dysfunction markers, cell-associated ICAM-I and VCAM-I play a role in the slow rolling and firm adhesion of monocytes to the arterial endothelium and initiation of atherosclerosis [52,53]. The upregulation of these adhesion receptors by IL-32 isoforms suggests a potential role for IL-32 in recruitment of immune cells and the contribution to the pathogenesis of atherosclerosis. Indeed, not only IL-32 isoforms upregulate the adhesion receptors but also increase the expression of a number of chemoattractants, namely, CCL-2, CXCL-1 and CXCL-8 at the RNA and protein levels. CCL-2, the prototype of chemoattractants, and CXCL-8 are both known to be strong mediators of monocyte and macrophage chemotaxis/recruitment, firm-adhesion and infiltration to the inflammation site [12,54]. The induction of these chemokines further highlights the potential role of IL-32 in the pathogenesis of atherosclerosis given the high expression of this cytokine in the atherosclerotic lesions [19]. Indeed, under the in vitro conditions that we employed in the current study, the IL-32-mediated expression of both CCL-2 and CXCL-8 by the coronary artery endothelial cells was sufficient to induce monocytes’ recruitment in a specific manner demonstrated by the CCR2 and CXCR2 antagonists (ligands of CCL-2 and CXCL-8, respectively). Monocytes play a crucial role in atherosclerosis by infiltrating the atherosclerotic lesions, differentiating into macrophages and forming inflammatory foam cells [55,56,57]. Given these effects, we suggest an inflammatory role for IL-32β and γ on endothelial cells leading to their dysfunction and release of both inflammatory cytokines and chemokines, which might promote arterial inflammation and monocyte recruitment to the atherosclerotic sites. The deleterious effects of chronic upregulation of IL-32 and IL-32-mediated chemokine expression may also expand beyond atherosclerosis. For instance, CCL-2 was recently shown to mediate early seeding of the HIV reservoir by recruiting a unique subset of CCR2/5+ CD4+ T-cells which become infected and form a significant reservoir for latent infection [58]. Similar studies were also reported on CXCL-1 (an IL-32 induced chemokine) on the enhancement of HIV replication [59]. Moreover, CXCL-8, another IL-32-induced chemokine, was shown to be upregulated in the plasma, serum and brain of PLWH presenting neurocognitive impairment [60].

While the current study as well as studies by other groups highlighted the role of IL-32 isoforms β and γ as proinflammatory [61], the role of IL-32α is not clearly established in the literature. In previous studies, the anti-atherosclerotic potential of IL-32α was demonstrated as it inhibits endothelial inflammation and vascular smooth muscle cell activation [62]. In line with these observations, our earlier studies suggested an anti-inflammatory role of IL-32α as it induces IL-10 expression in activated T-cells [22,24]. In the current study we further observed a tendency for IL-32α to decrease the production/secretion of ICAM-I and CXCL-8 compared to the other IL-32 isoforms. However, this effect did not reach statistical significance under the current experimental conditions. While our study was focused on the effect of exogenous IL-32 that mimics the circulating IL-32 proteins, it was also shown that intracellular IL-32 could induce the production of ICAM-I, IL-6 and CXCL-8 in HUVEC endothelial cells in response to IL-1β [63]. However, in the current study, IL-32 mediated these effects without the need for IL-1β, which suggests the independent role of IL-32 on endothelial inflammation and dysfunction.

The arterial endothelium plays a critical role in maintaining a healthy vascular tone [64]. Changes in the endothelium functions may lead to arterial stiffness [65,66] and contribute to CVD and heart failure (HF) [67]. On the other hand, IL-32 was recently reported to be upregulated in HF, with higher levels of IL-32 upon initial myocardial infarction predicting lower probability of HF-free status for a period of 2 years [68]. This aligns with our own observations on the negative association between the expression of IL-32 isoforms in blood from either PLWH or controls and the common carotid artery health marker cumulated lateral translation. Of note, we have recently shown that PLWH, when compared with controls, have lower lateral translation of common and internal carotid artery walls, measured with ultrasound elastography, which indicates increased vessel wall stiffness [32]. This was further accompanied with increased prevalence of carotid artery atherosclerotic plaques [32]. The negative associations between IL-32 isoforms and the arterial lateral translation further support the deleterious role of this inflammatory cytokine on the cardiovascular system. Meanwhile, this association was only observed in the relatively younger individuals of PLWH compared to the older population, which suggests that other inflammatory mediators may also play similar roles during the aging process. For instance, earlier studies demonstrated that inflammatory factors such as IL-6, hsCRP and D-dimers tend to remain higher in PLWH, even after HIV-RNA is suppressed by therapy [69] and that cytokines such as IL-6 are increased with aging [70]. Under this complex inflammatory condition, which characterizes the aging process of PLWH compared to the general population [71], it might be difficult to observe clear associations with and discern the disease progression to one single inflammatory mediator such as IL-32. 

One limitation for the current study was the failure to integrate sex and gender into our analysis of association between IL-32 expression and arterial stiffness. However, this was not possible as the carotid artery stiffness measures were carried out on participants from the Canadian HIV and Aging Cohort Study (CHACS), which is mainly a cohort of men with a very limited number of women participants [27]. Of note are our recent studies on two independent cohorts, the CHACS and the Women Interagency HIV Study (WIHS Cohort), which demonstrated that CVD-associated IL-32 isoforms are differentially expressed in men versus women living with HIV [25,26]. Therefore, future studies are warranted to explore these differences in larger cohorts integrating both men and women.

Another limitation of the current study is that we only tested three IL-32 isoforms at the functional level (IL-32α, β and γ) out of the 10 known IL-32 isoforms (α, β, γ, δ, D, ε, θ, ζ, η, and small/sm) [21,22], as these were the only commercially available isoforms. It thus remains unclear whether the rest of the IL-32 isoforms may complement, counteract, or simply play a redundant role on endothelial functions and carotid artery stiffening. Of note, our recent studies demonstrated that both IL-32α and β are upregulated in peripheral blood mononuclear cells in women living with HIV and are associated with carotid artery subclinical atherosclerosis [25]. While IL-32α seems to play an anti-inflammatory function by inducing IL-10 expression, IL-32β is highly inflammatory and induces multiple proinflammatory cytokines such as IL-6, TNF-α IL-1β, and IL-18 in T-cells and monocytes [21,26]. However, IL-32β represented the dominantly expressed isoform in individuals with subclinical atherosclerosis, which suggests that IL-32β-associated functions are the prevalent ones, at least in blood [25,26]. In line with our data, earlier studies also demonstrated that IL-32 is upregulated in the coronary artery endothelium from individuals with coronary artery disease, and that both IL-32β and γ are the dominantly expressed isoforms in the atherosclerotic arterial vessel wall [19,72]. Therefore, we believe that the functional data on IL-32β and γ presented in the current study are of clinical relevance and suggests a model in which persistent upregulation of IL-32 isoforms in PLWH is directly linked with vascular endothelial dysfunction and continuous recruitment of monocytes and other leukocytes, likely through IL-6-mediated mechanisms. Therefore, IL-32 may represent a potential therapeutic target to limit CVD in PLWH.

## Figures and Tables

**Figure 1 viruses-15-00700-f001:**
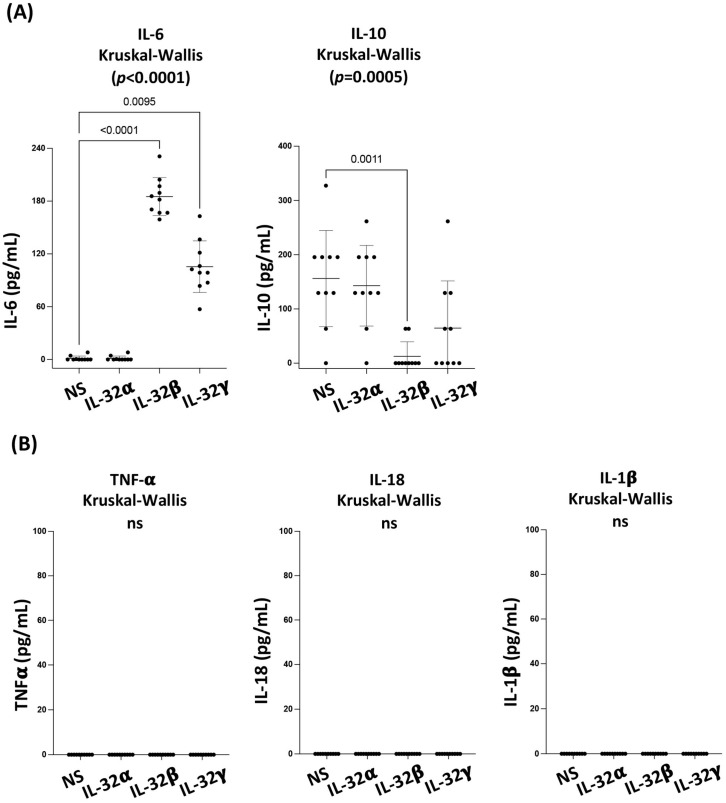
IL-32 isoforms exhibit differential impact on coronary artery endothelial cell’s cytokine production. (**A**) Modulation of IL-6 and IL-10 proteins in supernatants from CAEC following stimulation with IL-32α, β and γ for 72 h (500 ng/mL, n = 10). (**B**) TNF-α, IL-18 and IL-1β proteins in supernatants from CAEC following stimulation with IL-32α, β and γ for 72 h (500 ng/mL, n = 10). Dots on the graphs represent experimental replicates from CAEC. Data analyzed with the non-parametric test Kruskal-Wallis and Dunn’s subtest. NS: non-stimulated. ns: non-significant.

**Figure 2 viruses-15-00700-f002:**
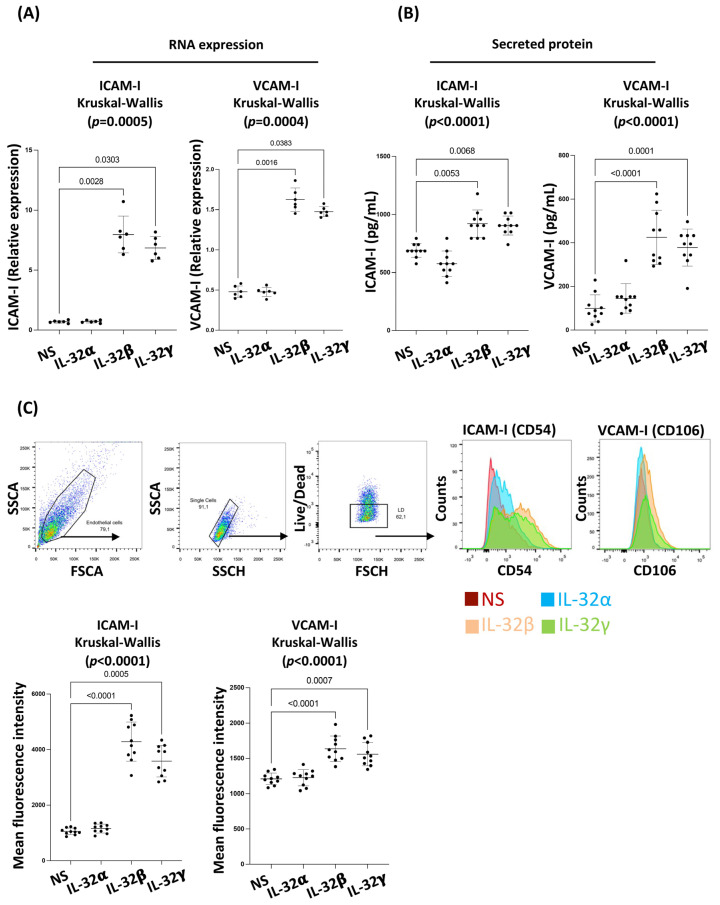
Coronary artery endothelial cell dysfunction markers are upregulated by IL-32 isoforms β and γ but not α. (**A**) Analysis of ICAM-I (CD54) and VCAM-I (CD106) RNA expression, normalized to the housekeeping gene β-glucuronidase, in primary CAEC showing their increased relative expression following 12 h stimulation with IL-32β and γ but not α (500 ng/mL, n = 6). (**B**) Analysis of ICAM-I and VCAM-I expression in CAEC supernatants following stimulation with IL-32α, β and γ for 72 h (n = 10). (**C**) Upper panels: Representative Flow cytometry data showing the gating strategy on CAEC live cells and overlapping histograms for ICAM-I and VCAM-I on the cell surface following stimulation with IL-32α, β and γ for 72 h (500 ng/mL). Lower panels: Analysis of ICAM-I and VCAM-I surface expression from n = 10 treatments (10 experimental replicates from CAEC). Data analyzed with the nonparametric test Kruskal-Wallis and Dunn’s subtest. NS: non-stimulated.

**Figure 3 viruses-15-00700-f003:**
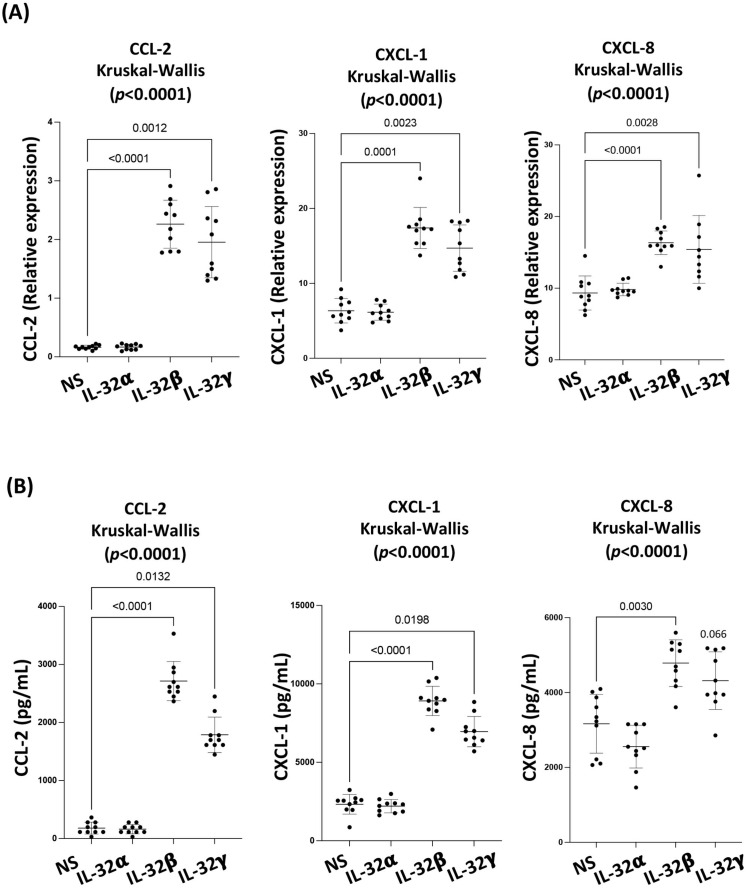
IL-32 β and γ increase chemokine production in coronary artery endothelial cells. (**A**) Analysis of CCL-2, CXCL-1 and CXCL-8 RNA expression, normalized to the housekeeping gene β-glucuronidase, in CAEC following 12 h stimulation with IL-32α, β and γ (500 ng/mL, n = 10). (**B**) Analysis for the production of CCL-2, CXCL-1 and CXCL-8 proteins in CAEC supernatants following stimulation with IL-32α, β and γ for 72 h (500 ng/mL, n = 10 experimental replicates from CAEC). Data analyzed with the non-parametric test Kruskal-Wallis and Dunn’s subtest. NS: non-stimulated.

**Figure 4 viruses-15-00700-f004:**
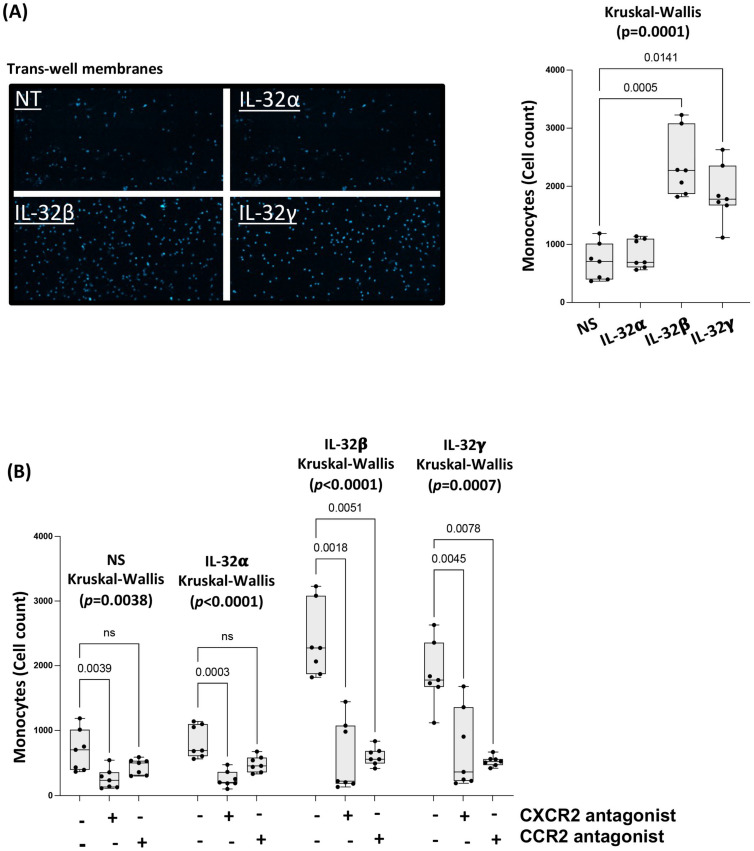
IL-32 β- and γ-induced chemokines drive monocyte transmigration toward CAEC supernatants. (**A**) Left panels: Representative fluorescent microscope images for monocytes attached to the lower side (surface facing the lower chambers) of the transwell membrane. Nuclei stained with DAPI on the transwell membranes. Right panel: Analysis for the numbers of migrated cells towards the CAEC supernatant stimulated with IL-32α, β and γ (n = 7). (**B**) Analysis for the impact of pre-treating monocytes with the antagonists of either CCR2 or CXCR2 prior to exposure to supernatants from CAEC non-stimulated or stimulated with the IL-32α, β and γ isoforms (n = 7). Dots on the graphs represent experimental replicates from CAEC. NS: non-stimulated. ns: non-significant.

**Figure 5 viruses-15-00700-f005:**
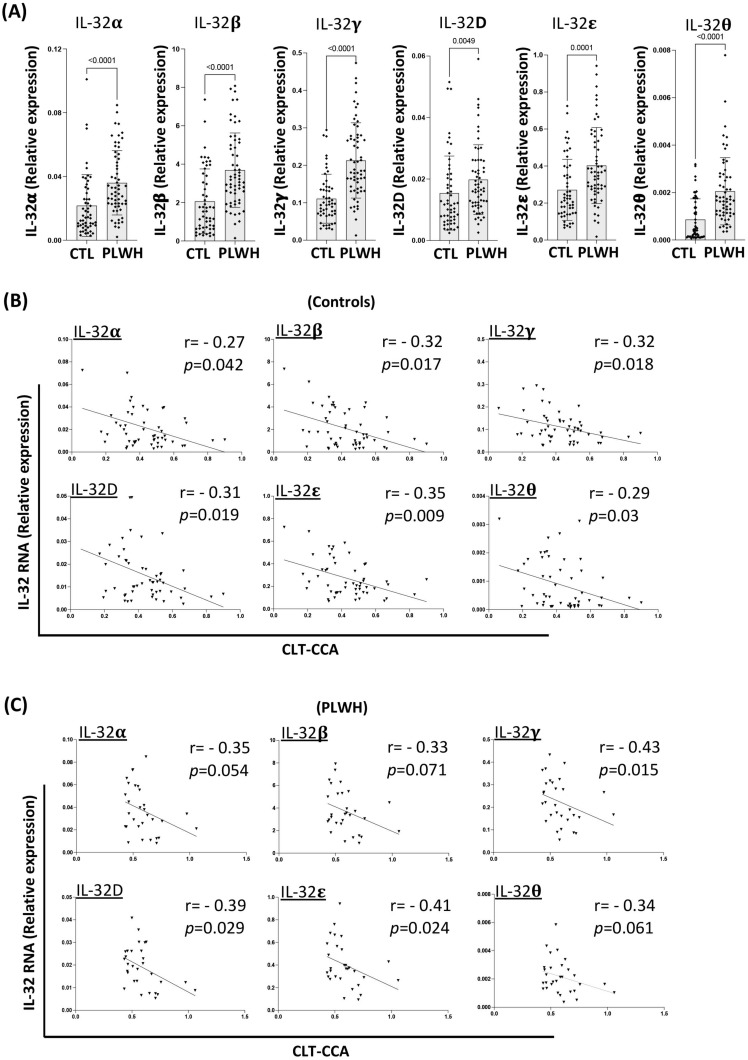
Association between IL-32 expression and carotid artery wall stiffness. (**A**) Increased expression of IL-32 isoforms in PBMCs from PLWH compared to controls (n = 60 and n = 53, respectively). IL-32 RNA levels were normalized to the housekeeping gene β-glucuronidase. (**B**) Correlation between IL-32 expression and carotid artery cumulative lateral translation (CLT) within the common carotid artery (CCA) from controls (n = 53). (**C**) Same as in B from PLWH participants (shown is association between IL-32 and CLT data from the upper quartiles, n = 30). Dots on the graphs represent data from individual participants in the CHACS cohort. CLT: Cumulated lateral translation in common carotid artery. CTL: Controls.

**Table 1 viruses-15-00700-t001:** Demographic and clinical parameters of study participants. Numbers are shown in Mean ± SD. N/A: non-applicable. NA: Non-available.

Variable	Controls	PLWH	*p* Value
Number of participants (Female/Male) Age (Years)	53 (5/48) 55.9 ± 8.32	60 (1/59) 57.6 ± 7.63	NS
Predicted 10 years Framingham Risk score ^(number of individuals with available data)^	11.08 ± 4.63 (50/53)	11.12 ± 6.77 (58/60)	NS
D-dimer (mg/L) ^(number of individuals with available data)^	0.301 ± 0.107 (18/53)	0.292 ± 0.155 (39/60)	NS
Body Mass Index (BMI) ^(number of individuals with available data)^	27.11 ± 4.83 (17/53)	24.98 ± 4.49 (48/60)	0.034
LDL–C (mmol/L) ^(number of individuals with available data)^	3.16 ± 0.77 (51/53)	2.86 ± 1.06 (55/60)	0.017
HDL–C (mmol/L) ^(number of individuals with available data)^	1.38 ± 0.39 (53/53)	1.24 ± 0.33 (58/60)	0.056 (NS)
Duration of infection (Years)	N/A	17.68 ± 7.9	
Duration of ART (Years)	N/A	14.36 ± 6.8	
Viral load (Log_10_ copies/mL)	N/A	1.6 ± 0.01	
Nadir CD4 count (cells/mm^3^)	N/A	215 ± 161	
CD4 count (cells/mm^3^)	NA	595 ± 228	
CD4/CD8 ratio	NA	0.9 ± 0.43	

## Data Availability

All reported data are included in the manuscript.

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
