# Peer review of "Differential Impact of IL-32 Isoforms on the Functions of Coronary Artery Endothelial Cells: A Potential Link with Arterial Stiffness and Atherosclerosis"

_viruses, 2023, doi:10.3390/v15030700_

Round 1
Reviewer 1 Report
In this manuscript, Bunet and colleagues build on their work investigating the roles of IL-32 isoforms in people with HIV (PLWH) and in atherosclerosis. Here, they treat coronary artery endothelial cells (CAECs) with IL-32 alpha, beta, and gamma isoforms and find that the beta- and gamma-treated CAECs produce IL-6, ICAM-1, V-VCAM-1, and the chemokines CCL2, CXCL1, ad CXCL8. Furthermore, treatment also seemed to reduce CAEC production of the anti-inflammatory cytokine IL-10. In these in vitro studies, IL-32 alpha serves mainly as a negative control, as it does not induce similar responses as the other isoforms. The authors then use a transwell system to investigate the recruitment of monocytes by IL-32-treated CAECs; this recruitment is at least partially dependent on the CCR2 and CXCR2 chemokine receptors. Finally, they investigate the association of IL-32 with arterial stiffness (as measured by cumulative lateral translation within the common carotid artery [CLT-CCA]) in people with and without HIV. Plasmas from PLWH had elevated levels of RNAs from all IL-32 isoforms compared to plasmas from control donors, and in both groups, IL-32 RNA levels were significantly positively associated with age. Among control participants, these levels were also all significantly negatively correlated with CLT-CCA (and thus positively correlated with arterial stiffness). Among PLWH, however, there was no association of IL-32 RNA levels with arterial stiffness when the full population (n=60) was tested. When only those PLWH in the lowest CLT-CCA quartiles (most arterial stiffness) were assessed, each IL-32 RNA was significantly negatively associated with CLT-CCA, which is somewhat surprising for IL-32alpha, given its lack of proinflammatory activity in their earlier assays. This study is comprehensive and well-written and provides insight into a potential mechanism of elevated cardiovascular risk among PLWH.
Specific comments
1. The authors should strive to use person-first language as much as possible and avoid terms like “HIV-infected” or “HIV+”. This goes for the controls, too, who are often referred to as “HIV-uninfected”.
2. For the figures that use CAECs, do the N values come from unique donors, or from different batches (or thaws) of cells from the same donor?
3. The assays in Figure 1B would benefit from a positive control treatment, maybe LPS or some other TLR ligand?, to show that the CAECs are even capable of producing TNFa, IL-18, and IL-1b.
4. I’m a bit confused by the transwell assay. The authors describe it as an assay to measure monocyte transmigration toward CAECs in the text, but it’s hard to discern that is the case from the methods. For instance, are there actual CAECs in the assay, or are the cells migrating toward culture supernatants? In my reading (and I could be wrong) it seems like there are no CAECs in the assay. If just supernatants, the legend should say “…toward CAEC supernatants” instead of “…toward CAEC.” Furthermore, while IL-32 alpha does not enhance monocyte recruitment over no treatment controls or induce CXCL8 expression from the CAECs, CXCR2 antagonism (which blocks CXCL8 activity) inhibits the number of monocytes that traffic in response to IL-32 alpha. Why? Did the authors use these inhibitors in the untreated supernatant assays?
5. I recommend the authors describe the transwell assay in more detail in both the methods and the results and be very careful in their interpretation of it.
6. The figure legend for supplementary figure 1 does not match the actual figure.
Author Response
We thank the reviewer for stating that our study is comprehensive and provides insight into the potential mechanisms of elevated cardiovascular risk in PLWH.
Reviewer: The authors should strive to use person-first language as much as possible and avoid terms like “HIV-infected” or “HIV+”. This goes for the controls, too, who are often referred to as “HIV-uninfected”.
Answer: We agree with the reviewer that person-first language should always be used. We have gone through the text and figures and figure legends to modify the term HIV+ population/individuals to people living with HIV (PLWH) and also to modify the HIVneg controls to only “controls”.
Reviewer: For the figures that use CAECs, do the N values come from unique donors, or from different batches (or thaws) of cells from the same donor?
Answer: In fact, we are using primary cells obtained from human coronary artery tissues through a commercial source (Creative BioArray) isolated from one single donor. Therefore, the “n” in the different in vitro assays represents experimental replicates using the same cells. We have now clarified this in the Figure legends (Figures 1 through Figure 4). In Figure 5, the “n” represents data from individual participants in the study.
Reviewer: The assays in Figure 1B would benefit from a positive control treatment, maybe LPS or some other TLR ligand?, to show that the CAECs are even capable of producing TNFa, IL-18, and IL-1b.
Answer: This is an excellent suggestion. In response to the reviewer’s comment, we re-tested TNFa, IL-1b and IL-18 by ELISA from stored supernatants of CAEC treated with either IL-32 isoforms or LPS and we could not detect any quantifiable levels for these three cytokines, even with LPS. These results suggest that the primary CAEC are not producers for these cytokines. We have now added this information to the results and discussion sections (pages 4, lines 241-242 and page 13, lines 475-476).
Reviewer: I’m a bit confused by the transwell assay. The authors describe it as an assay to measure monocyte transmigration toward CAECs in the text, but it’s hard to discern that is the case from the methods. For instance, are there actual CAECs in the assay, or are the cells migrating toward culture supernatants? In my reading (and I could be wrong) it seems like there are no CAECs in the assay. If just supernatants, the legend should say “…toward CAEC supernatants” instead of “…toward CAEC.” Furthermore, while IL-32 alpha does not enhance monocyte recruitment over no treatment controls or induce CXCL8 expression from the CAECs, CXCR2 antagonism (which blocks CXCL8 activity) inhibits the number of monocytes that traffic in response to IL-32 alpha. Why? Did the authors use these inhibitors in the untreated supernatant assays?
Answer: We thank the reviewer to bring this point to our attention. In fact, we used supernatant from CAEC cells stimulated with the different IL-32 isoforms in all the transmigration assays. As suggested by the reviewer, we have now made it clear in the text and the figure legend that only CAEC supernatants were used.
Reviewer: Furthermore, while IL-32 alpha does not enhance monocyte recruitment over no treatment controls or induce CXCL8 expression from the CAECs, CXCR2 antagonism (which blocks CXCL8 activity) inhibits the number of monocytes that traffic in response to IL-32 alpha. Why? Did the authors use these inhibitors in the untreated supernatant assays?
Answer: This is true that under conditions of IL-32alpha stimulation, where IL-32alpha does not induce neither CCL2 nor CXCL-8, we still observe a negative impact for the CXCR2 antagonism. This is likely due to the inhibition of the spontaneous expression of CXCL-8 and CXCL-1 (both share the CXCR2 receptor) by the endothelial cells (please see Figure 3B, middle and right panels). This might be further supported by the fact that we also observed a significant level of inhibition in the non-treated conditions (no IL-32) when CXCR2 antagonist was used. We have now modified Figure 4 to show the inhibitory effect of CXCR2 antagonist on monocyte migration in nontreated cells from the same experiments and modified the text in both the results section and figure legend.
Reviewer: I recommend the authors describe the transwell assay in more detail in both the methods and the results and be very careful in their interpretation of it.
Answer: Thanks for the recommendation, we have now added more details in the Materials and Methods section, page 3, lines 183-185.
Reviewer: The figure legend for supplementary figure 1 does not match the actual figure.
Answer: Thanks to bring this to our attention. We corrected the Figure Legend.
Reviewer 2 Report
In this study Bunet and colleagues build on their previous work to investigate the pro-atherogenic effects of IL-32 isoforms on human coronary artery endothelial cells (HCAECS), with relevance to people living with HIV. They show that treatment of CAECS with IL-32β and γ in vitro induces elevated IL-6, ICAM-1, VCAM-1, CCL2 etc gene expression and protein production and promotes monocyte transmigration. This is relevant to people with HIV as PBMCs from HIV+ individuals had an elevated level of all IL-32 isoforms. Interestingly, levels of IL-32 isoforms only correlated with surrogate markers of coronary artery disease in younger HIV+ individuals with more significant arterial stiffness, suggesting an age-related mediator was more important in atherogenesis in older PWH. Together, the manuscript is well written and provides evidence for an important role of IL-32 isoforms (particularly β and γ) in the pathogenesis of atherosclerosis in younger people living with HIV.
Comments:
· It is interesting that HIV-seronegative individuals show negative correlations between IL-32 isoforms and arterial stiffness, whilst this is only present in younger PWH with CLT-CCA scores in the upper quartiles suggesting an increased age-related effect in PWH. Can the authors speculate further as to the reasoning behind this in the discussion?
· - Given that IL-32 gene expression levels increased with age and that PWH with worse arterial stiffness were older, were linear regression models adjusting for age and other traditional risk factors including LDL, smoking status etc. performed? This approach may help to delineate the associations with IL-32 that are not affected by these confounders.
· All stimulations were performed in the presence of polymyxin B to inhibit possible LPS contamination. Were polymyxin untreated conditions also performed to confirm this treatment did not impact cell function or response to IL-32 isoforms?
· Fig 2 – It may be of interest to further validate elevated ICAM-1 protein levels on cells using quantitative immunofluorescence analysis as a large proportion of cells (~40%) appear to have died post-accutase treatment (fig 2C).
Author Response
Reviewer: It is interesting that HIV-seronegative individuals show negative correlations between IL-32 isoforms and arterial stiffness, whilst this is only present in younger PWH with CLT-CCA scores in the upper quartiles suggesting an increased age-related effect in PWH. Can the authors speculate further as to the reasoning behind this in the discussion?
Answer: We thank the reviewer for this comment; we have addressed this question in the discussion section as we believe that other inflammatory mediators may also play similar roles during the aging process. For instance, earlier studies demonstrated that inflammatory factors such as IL-6, hsCRP and D-dimers tend to remain higher in PLWH, even after HIV-RNA is suppressed by therapy and that cytokines such as IL-6 are increased with aging. Given the impact of IL-6 on arterial functions and CVD, it is then very likely that a complex of inflammatory mediators may cooperate in arterial stiffness in older aging. We have now modified the discussion section (Page 15, lines 577-585) to reflect this discussion.
Reviewer: Given that IL-32 gene expression levels increased with age and that PWH with worse arterial stiffness were older, were linear regression models adjusting for age and other traditional risk factors including LDL, smoking status etc. performed? This approach may help to delineate the associations with IL-32 that are not affected by these confounders.
Answer: Thanks for the suggestion. We ran the linear regression analyses for IL-32 correlations with arterial stiffness in both controls and PLWH and integrated the outcomes to the results section (page xx, lines xx). Indeed, in PLWH within the upper CLT-CCA quartiles, upon adjusting for age, Framingham risk score, body mass index, LDL and HDL cholesterol levels, the associations between IL-32 and CLT-CCA strengthened and were statistically significant for all isoforms except for θ (p=0.09) (page 11, lines 407-419).
Reviewer: All stimulations were performed in the presence of polymyxin B to inhibit possible LPS contamination. Were polymyxin untreated conditions also performed to confirm this treatment did not impact cell function or response to IL-32 isoforms?
Answer: In fact, yes, during the optimization process, we have tested the impact of polymyxin B on CAEC in the presence or absence of IL-32 isoforms. We have now added a new Supplemental figure (Supplemental Figure 1) showing that in the presence of polymyxin B, IL-32-mediated production of chemokines was slightly decreased without reaching statistical significance. In contrast, polymyxin B completely diminished the LPS-mediated upregulation of the tested CXCL-1 chemokine. We included this in the results section (page 7, lines 311-315).
Reviewer: Fig 2 – It may be of interest to further validate elevated ICAM-1 protein levels on cells using quantitative immunofluorescence analysis as a large proportion of cells (~40%) appear to have died post-accutase treatment (fig 2C).
Answer: We agree with the reviewer that the accutase treatment may impact the viability of cells, however since it is a necessary step to get these cells in suspension for the flow cytometry assays, we made sure to gate only on the live cells. Furthermore, other groups “(Quan, Y. et al., 2019: Impact of cell dissociation on identification of breast cancer stem cells. Cancer Biomark. 12, 125–133 (2012) and Skog, M. et al, 2019: The effect of enzymatic digestion on cultured epithelial autografts. Cell Transplant. 28, 638–644 (2019))” showed that accutase is a mild acting enzyme that does not impact the expression of surface molecules. In addition, we have further confirmed the upregulation of ICAM-1 at the RNA level and also at the protein level by ELISA (Figure 2A&B).
Reviewer 3 Report
People living with HIV (PLWH) suffer from complications such as cardiovascular disease (CVD) despite successful antiviral treatment. Although chronic inflammation has been linked to such complications, the mechanisms underlying chronic inflammation and CVD in PLWH have remained elusive. In this manuscript, Bunet and colleagues investigated the role of IL-32 isotypes in regulating proinflammatory cytokine and chemokine production and adhesion molecule expression in coronary artery endothelial cells (CAEC), which have been linked to atherosclerosis. The authors show that IL-32β and IL-32γ induce the production of IL-6 and several chemokines and upregulate ICAM-I and VCAM-I expression in CAECs. Moreover, the authors claim that IL-32 expression in both PLWH and non-infected controls correlates with the carotid artery stiffness.
This report on the role of IL-32 in CVD in PLWH is interesting and offers insights that are potentially important to our general understanding of HIV-associated complications in PLWH. However, there are several points that need to be addressed in order to reveal the relevance of these findings. Some specific questions and concerns are the following.
Major points:
1. The authors’ conclusion that “IL-32 expression in both PLWH and non-infected controls correlates with the carotid artery stiffness, measured by the cumulated lateral translation” is not fully supported by the data. As shown in Supplemental Figure1, in PLWH, there was no significant correlation between IL-32 and CLT-CCA. The difference was significant only in PLWH with higher CLT-CCA.
2. Some ELISA data using CAECs look strange and need an explanation. For example, in Fig1A IL-10, there are four dots in NT and three dots in IL-32a around 200 pg/ml. It seems they are all aligned (same value). Is that coincidence? What does individual dot represent? If it represents a single data point from an independent experiment (please clarify this in the legend), it is unusual to have the same value many times. Also, primary CAECs were obtained from a commercial source. Are they from different donors? If so, it is even unlikely to have the same values. Similar data are seen in other figures like Fig2.
3. Data in Fig4A right panel and Fig4B (no antagonist condition: the leftmost column of each figure) look similar or same. If so, baseline migration (NT) in Fig4A should be applied to Fig4B. Then, a question is why one antagonist for either CXCR2 or CCR2 reduces monocyte migration to the NT level. Even if one is blocked, isn’t it likely that the other chemokine(s) be active? Have the authors tested purified chemokines (CCL2, CXCL1, CXCL8) for monocyte migration assay? It would be a nice positive control.
4. This cohort is male-oriented. Are there any gender-related differences known in IL-32 biology? It would be great if the authors could discuss this point.
Minor:
· Line 65-66, “elevated levels of the inflammatory cytokines IL-1β, IL-6 and TNF-α inflammatory cytokines contribute”. Typo?
Author Response
Major points:
Reviewer: The authors’ conclusion that “IL-32 expression in both PLWH and non-infected controls correlates with the carotid artery stiffness, measured by the cumulated lateral translation” is not fully supported by the data. As shown in Supplemental Figure1, in PLWH, there was no significant correlation between IL-32 and CLT-CCA. The difference was significant only in PLWH with higher CLT-CCA.
Answer: We agree with the reviewer that significant correlations between the expression of IL-32 isoforms and artery stiffness in PLWH, was only observed in participants with higher CLT-CCA and not in the total population. We have now modified the conclusions to stress this point and to further discuss the lack of associations in individuals with lower CLT-CCA (older individuals). We have added the following clarification in the discussion on page 15, lines 577-585: “Meanwhile, this association was only observed in the relatively younger individuals of PLWH compared to the older population, which suggests that other inflammatory mediators may also play similar roles during the aging process. For instance, earlier studies demonstrated that inflammatory factors such as IL-6, hsCRP and D-dimers tend to remain higher in PLWH, even after HIV-RNA is suppressed by therapy and that cytokines such as IL-6 are increased with aging. Under this complex inflammatory condition, which characterizes the aging process of PLWH compared to the general population, it might be difficult to observe clear associations with and discern the disease progression to one single inflammatory mediator such as IL-32”.
Reviewer: Some ELISA data using CAECs look strange and need an explanation. For example, in Fig1A IL-10, there are four dots in NT and three dots in IL-32a around 200 pg/ml. It seems they are all aligned (same value). Is that coincidence? What does individual dot represent? If it represents a single data point from an independent experiment (please clarify this in the legend), it is unusual to have the same value many times. Also, primary CAECs were obtained from a commercial source. Are they from different donors? If so, it is even unlikely to have the same values. Similar data are seen in other figures like Fig2.
Answer: Thanks for this observation. We have clarified in the Material and Methods section that the CAEC primary cells were obtained from a commercial source (Creative BioArray) and they are originating from one single donor. The individual dots on each figure for the in vitro data represent experimental replicates from these primary cells and not individual donors and this might in part explain the close values for some measures. However, for the IL-10 ELISA (and IL-6), we would like to highlight that the cytokine production by these primary CAEC was relatively modest and close to the detection limit. At this lower level of production, some of the ELISA readings (optical densities) for the IL-10 replicates were similar in some instances as specified by the reviewer. Yet, the production of these cytokines was still measurable by our optimized ELISA assays as per the manufacturer conditions where the ELISA readings fell within the lower limit of the standard curve (generated using the recombinant proteins provided by the supplier). We have attached here the original ELISA readings of IL-10 (low production), IL-18 (no production at all), and also CXCL-1 (high production) for comparison.
Reviewer: Data in Fig4A right panel and Fig4B (no antagonist condition: the leftmost column of each figure) look similar or same. If so, baseline migration (NT) in Fig4A should be applied to Fig4B.
Answer: This is exact, Figure 4A, shows the absolute numbers of migrated monocytes to the supernatants of CAEC treated with the three IL-32 isoforms. The goal of showing this panel is to compare the potential of each isoform to mediate monocytes chemoattraction. However, in Figure 4B, the same data from A were repeated but this time coupled with the effect of receptor antagonists for CCR2 and CXCR2 to demonstrate the specificity of the assay to the studied chemokines, CCL2 and CXCL-8. As suggested by the reviewer, we also moved the numbers of the non-treated (now we call them non-stimulated) cells to the new Figure 4B. We also included new data showing the numbers of migrated monocytes for the non-stimulated conditions in the presence or absence of receptor antagonists as even the non-stimulated cells (no IL-32) still produce significant levels of chemokines (please see Figures 3B) and are thus susceptible to both inhibitors of CCL-2 and CXCL-8 receptors.
Reviewer: Then, a question is why one antagonist for either CXCR2 or CCR2 reduces monocyte migration to the NT level. Even if one is blocked, isn’t it likely that the other chemokine(s) be active? Have the authors tested purified chemokines (CCL2, CXCL1, CXCL8) for monocyte migration assay? It would be a nice positive control.
We agree with the reviewer that other chemokine(s) should remain active in the migration assays upon the inhibition of one given chemokine. Indeed, earlier studies (please see Reference 1 below) [1] demonstrated that CXCL-8 has a modest chemoattraction potential compared to CCL-2, however, CCL-2-mediated chemoattraction of monocytes is largely dependent on CXCL-8 and that’s why these two chemokines are often co-expressed together in response to inflammatory cytokines. These studies demonstrated that CXCL-8 largely enhances the chemoattraction potential of CCL-2 through mechanisms involving the activation of ERK1/ERK2 and intracellular calcium[1]. Then inhibition of CCL-2 in our assays would likely diminish the monocytes' migration as expected but also CXCL-8 inhibition would on its turn diminish the CCL-2-mediated migration as well and to likely close levels. We have now explained this in the results section and cited the earlier studies (page 9, lines 357-367).
Reviewer: This cohort is male-oriented. Are there any gender-related differences known in IL-32 biology? It would be great if the authors could discuss this point.
Answer: Differences in IL-32 biology as related to sex and gender are yet to be studied. However, our recent studies on two independent cohorts; the Canadian HIV and Aging Cohort Studies (CHACS) and the Women Interagency HIV Study, (WIHS Cohort) demonstrated that CVD-associated IL-32 isoforms are differentially expressed in men versus women living with HIV [2, 3]. Even though, in the current study, we could not integrate sex and gender to our analysis of the association between IL-32 expression and arterial stiffness given that carotid artery stiffness measures were carried out on participants from only the CHACS cohort that is mainly men cohort with a very limited number of women participants [4]. Therefore, future studies are thus warranted to explore these differences in larger cohorts integrating both men and women. We have now discussed this further in the manuscript (page 15, lines 587-595).
Minor:
Reviewer: Line 65-66, “elevated levels of the inflammatory cytokines IL-1β, IL-6 and TNF-α inflammatory cytokines contribute”. Typo?
Answer: Thanks, we corrected the typo.
References
- Gouwy M, Struyf S, Noppen S, Schutyser E, Springael JY, Parmentier M, Proost P, Van Damme J. Synergy between coproduced CC and CXC chemokines in monocyte chemotaxis through receptor-mediated events. Mol Pharmacol 2008; 74(2):485-495.
- El-Far M, Durand M, Turcotte I, Larouche-Anctil E, Sylla M, Zaidan S, Chartrand-Lefebvre C, Bunet R, Ramani H, Sadouni M, Boldeanu I, Chamberland A, Lesage S, Baril JG, Trottier B, Thomas R, Gonzalez E, Filali-Mouhim A, Goulet JP, Martinson JA, Kassaye S, Karim R, Kizer JR, French AL, Gange SJ, Ancuta P, Routy JP, Hanna DB, Kaplan RC, Chomont N, Landay AL, Tremblay CL. Upregulated IL-32 Expression And Reduced Gut Short Chain Fatty Acid Caproic Acid in People Living With HIV With Subclinical Atherosclerosis. Front Immunol 2021; 12:664371.
- El-Far M, Hanna DB, Durand M, Larouche-Anctil E, Sylla M, Chartrand-Lefebvre C, Cloutier G, Goulet JP, Kassaye S, Karim R, Kizer JR, French AL, Gange SJ, Lazar JM, Hodis HN, Routy JP, Ancuta P, Chomont N, Landay AL, Kaplan RC, Tremblay CL. Brief Report: Subclinical Carotid Artery Atherosclerosis Is Associated With Increased Expression of Peripheral Blood IL-32 Isoforms Among Women Living With HIV. J Acquir Immune Defic Syndr 2021; 88(2):186-191.
- Durand M, Chartrand-Lefebvre C, Baril JG, Trottier S, Trottier B, Harris M, Walmsley S, Conway B, Wong A, Routy JP, Kovacs C, MacPherson PA, Monteith KM, Mansour S, Thanassoulis G, Abrahamowicz M, Zhu Z, Tsoukas C, Ancuta P, Bernard N, Tremblay CL, investigators of the Canadian HIV, Aging Cohort S. The Canadian HIV and aging cohort study - determinants of increased risk of cardio-vascular diseases in HIV-infected individuals: rationale and study protocol. BMC Infect Dis 2017; 17(1):611.

Round 2
Reviewer 3 Report
The authors have responded well to the concerns raised and significantly improved the clarity of the manuscript.